# Inhibitory Effects of 2-Aminoethoxydiphenyl Borate (2-APB) on Three K_V_1 Channel Currents

**DOI:** 10.3390/molecules28020871

**Published:** 2023-01-15

**Authors:** Wei Zhao, Lanying Pan, Antony Stalin, Jianwei Xu, Liren Wu, Xianfu Ke, Yuan Chen

**Affiliations:** 1Zhejiang Provincial Key Laboratory of Resources Protection and Innovation of Traditional Chinese Medicine, Zhejiang Agriculture and Forestry University, Hangzhou 311300, China; 2The State Key Laboratory of Subtropical Silviculture, Zhejiang Agriculture and Forestry University, Hangzhou 311300, China; 3Institute of Fundamental and Frontier Sciences, University of Electronic Science and Technology of China, Chengdu 610054, China; 4Zhejiang Key Laboratory for Laboratory Animal and Safety Research, Hangzhou Medical College, Hangzhou 311300, China

**Keywords:** 2-APB, K_V_1.2, K_V_1.3, K_V_1.4, potassium channel

## Abstract

2-Aminoethoxydiphenyl borate (2-APB), a boron-containing compound, is a multitarget compound with potential as a drug precursor and exerts various effects in systems of the human body. Ion channels are among the reported targets of 2-APB. The effects of 2-APB on voltage-gated potassium channels (K_V_) have been reported, but the types of K_V_ channels that 2-APB inhibits and the inhibitory mechanism remain unknown. In this paper, we discovered that 2-APB acted as an inhibitor of three representative human K_V_1 channels. 2-APB significantly blocked A-type Kv channel K_V_1.4 in a concentration-dependent manner, with an IC_50_ of 67.3 μM, while it inhibited the delayed outward rectifier channels K_V_1.2 and K_V_1.3, with IC_50_s of 310.4 μM and 454.9 μM, respectively. Further studies on K_V_1.4 showed that V549, T551, A553, and L554 at the cavity region and N-terminal played significant roles in 2-APB’s effects on the K_V_1.4 channel. The results also indicated the importance of fast inactivation gating in determining the different effects of 2-APB on three channels. Interestingly, a current facilitation phenomenon by a short prepulse after 2-APB application was discovered for the first time. The docked modeling revealed that 2-APB could form hydrogen bonds with different sites in the cavity region of three channels, and the inhibition constants showed a similar trend to the experimental results. These findings revealed new molecular targets of 2-APB and demonstrated that 2-APB’s effects on K_V_1 channels might be part of the reason for the diverse bioactivities of 2-APB in the human body and in animal models of human disease.

## 1. Introduction

Boron, often as boric acid, is a trace and necessary element for bone metabolism, inflammation attenuation, wound healing, modulating hormone levels, enhancing magnesium absorption, attenuating oxidative stress, and ameliorating cisplatin-induced peripheral neuropathy. 2-Aminoethoxydiphenyl borate (2-APB), a boron-containing compound, exerts effects on neurons, adaptive and innate immunity, and muscle cells, and it provides a cytoprotective effect against reactive oxygen. The molecular pharmacology of 2-APB reveals its complex actions through multiple targets including ion channels, transporters, and enzymes. 2-APB has the potential to act as both as a drug and as a precursor of drugs. Interest in 2-APB and its derivatives regarding the design of bioactive molecules keeps increasing in recent years [1,2].

The ion channel is the third-largest family of signaling molecules and the second-largest druggable target group. 2-APB has been shown to act as either a blocker or an activator for various ion channels and is thus extensively used as a modulator of different types of ion channels in many studies [2,3,4,5]. As for potassium channels, 2-APB potently inhibited K_Ca_3.1 in a reversible manner under a whole-cell patch clamp (half maximal inhibition 14.2 μM) [6]. 2-APB was found to be an effective activator for all members of the TREK subfamily (hKCNK2, hKCNK4, and hKCNK10), with the highest efficacy in hKCNK10 [3]. The effects of 2-APB on voltage-gated potassium channels (K_V_) have also been reported. 2-APB at 100 μM reversibly inhibited both the transient and sustained voltage-activated potassium current of *Limulus ventral* photoreceptors during depolarizing steps [7]. 2-APB also inhibited the delayed rectifier K^+^ current (I_K_), with an IC_50_ of 120 μM in guinea pig arteriole cells [8]. However, the types of K_V_ channels inhibited and the inhibitory mechanism remain unknown.

Voltage-gated potassium channels mainly respond in the repolarization of action potentials and participate in the regulation of resting potentials in non-excitable cells involved in many physiological processes [9]. K_V_1, the largest subfamily of K_V_ channels, includes at least eight genes (KCNA1-8) that encode K_V_1.1–K_V_1.8 proteins. A K_V_1 channel is formed by four α-subunits encircling a central ion conduction pore. Like many other K_V_ channels, each subunit of K_V_1 consists of six transmembrane segments (S1 to S6) with a membrane re-entering P-loop (loop of S5–S6). Functional K_V_1 channels are widely expressed throughout the nervous system and are found in peripheral tissues such as the heart, the vasculature, and the immune system [10,11]. Although K_V_1 channels are conserved in a protein sequence, they can be divided into two types by their current characteristics. K_V_1.4 is a typical transient A-type channel, while the other seven types, including K_V_1.2 and K_V_1.3, are delayed outward rectification channels [12]. In this study, we discovered that 2-APB could inhibit the two types of K_V_1 currents with different potencies, and we studied the potential inhibitory mechanism.

## 2. Results

### 2.1. 2-APB Effects on Three hK_V_1 Channel Currents

We investigated 2-APB’s effects on the K_V_1.2, K_V_1.3, and K_V_1.4 channel currents.

K_V_1.2 currents were evoked in stably transfected CHO cell lines by a 2000 ms pulse at 20 mV from a holding potential at −80 mV (Figure 1A). Cumulative 2-APB concentrations (10, 30, 100, 300, and 1000 μM) were perfused directly to the cells by gravity. Data were obtained once the responses to 2-APB reached a steady state. The maximal current amplitudes were measured at the end of pulses. The percentage inhibitions by 10, 30, 100, 300, and 1000 μM 2-APB were 3.5 ± 1.2%, 12.8 ± 2.3%, 22.9 ± 2.7%, 43.4 ± 2.6%, and 77.8 ± 1.0%, respectively (*n* = 4; Figure 1B,C). The IC_50_ value was 310.4 ± 29.4 μM (*n* = 9). To assess the effect of 2-APB on the current–voltage (I-V) relationship, we constructed I-V curves with and without 2-APB in the bath (100 μM). 2-APB made the I-V curves shift to the right (Figure 1D).

K_V_1.3 currents were evoked by a 2000 ms length and +40 mV pulse from a holding potential at −80 mV. 2-APB conspicuously blocked K_V_1.3 currents (Figure 2A). Furthermore, 10, 30, 100, 300, and 1000 μM 2-APB was cumulatively perfused directly onto the cells, and it inhibited peak amplitudes by 7.2 ± 1.7%, 16.6 ± 2.9%, 25.3 ± 5.2%, 36.9 ± 4.2%, and 71.4 ± 5.8%, respectively (*n* = 6) (Figure 2B,C). The IC_50_ value was 454.9 ± 120.9 μM (*n* = 7). The I-V relationship curves before and after 2-APB application (100 μM) showed that the voltage dependence of peak currents apparently shifted (Figure 2D).

The concentration–response curves of K_V_1.2 and K_V_1.3 did not seem sigmoidal due to the responses not reaching the maximum yet.

K_V_1.4 currents were elicited with a voltage step (500 ms duration) to −10 mV from a holding potential of −80 mV. The currents evoked by this voltage-clamp protocol were transient A-type K^+^ currents that decayed rapidly. The percentage inhibitions by 10, 30, 100, 300, and 1000 μM 2-APB were 16.9 ± 1.6%, 30.3 ± 3.5%, 56.8 ± 2.7%, 77.9 ± 3.0%, and 93.8 ± 1.7%, respectively (*n* = 6) (Figure 3A–C). The IC_50_ value and Hill coefficient were 67.3 ± 5.32 μM (*n* = 6) and 1.59 ± 0.13, respectively. The I-V relationship studies showed that Kv1.4 currents were elicited by voltage pulses more positive than −40 mV, and the current amplitude increased linearly with further depolarization. The presence of 2-APB reduced the current amplitude over the entire voltage range, which activated the currents (Figure 3D).

### 2.2. Characteristics of the Inhibition of 2-APB on Kv1.4

To understand the potential inhibition mechanisms of 2-APB, we further investigated the 2-APB effect on K_V_1.4 since 2-APB was the most potent in inhibiting Kv1.4 among three K_V_1 channels. A two-pulse protocol (Figure 4A) was set to examine the recovery of the inactivation. Currents were evoked by the same amplitude, a 40 mV prepulse, and a test pulse from a holding potential at −80 mV. The interval between these two pulses was varied, and both peak currents evoked from the test pulse (I_2_) and the prepulse (I_1_) were recorded. The ratio of I_2_/I_1_ was plotted against the intervals and fitted by a single exponential. The time constant was 683.0 ± 80.5 ms and 624.2 ± 78.8 ms before and after 2-APB (100 μM) application (*n* = 5, *p* > 0.05) (Figure 4B). These data indicate that 2-APB had no effects on the recovery of the inactivation.

The effects on the onset of inactivation by 2-APB were also investigated. A two-pulse protocol (Figure 4A) was used. Currents were evoked by the same amplitude, a 40 mV prepulse, and a test pulse from a holding potential at −80 mV. The duration of the prepulse was varied, while the interval between these two pulses was fixed at 20 ms. Both peak currents evoked from the test pulse (I_2_) and the prepulse (I_1_) were recorded. The ratio of I_2_/I_1_ was plotted against the intervals and fitted by a single exponential. The onset of slow inactivation was measured by a prepulse duration from 32 ms to 2048 ms. The time constant was 74.5 ± 11.5 ms and 67.0 ± 8.2 ms before and after 2-APB (100 μM) application (*n* = 4, *p* > 0.05) (Figure 4D). The onset of fast inactivation was measured by a prepulse duration from 2 ms to 120 ms. However, the fit started at 8 ms. The time constancy was 36.2 ± 2.1 ms before and was 32.9 ± 0.9 ms after 2-APB (100 μM) application (*n* = 5, *p* > 0.05) (Figure 4C). These findings show that 2-APB had no effects on the onset of inactivation.

Figure 4C illustrates that the 2 ms length prepulse facilitated currents evoked by the test pulse. The facilitation of the amplitudes of currents (I_2_/I_1_) was 1.21 ± 0.03 vs. 2.80 ± 0.32 before and after 2-APB application (*n* = 5). It seemed like the facilitation was more than doubled after 2-APB application. However, it was actually because 2-APB suppressed the currents of the prepulse (I_1_) more than those of the test pulse (I_2_, Figure 5A,B). At this stage, the kinetics of the activation of the current were well fitted by the Boltzmann equation rather than a single exponential (Figure 5C). The average time to activate half of the maximal current was delayed by 0.6 ± 0.15 ms by 30 μM 2-APB (*p* < 0.01, *n* = 10) (Figure 5D). Meanwhile, the slope, which measured the speed of the activation, was also significantly slowed (*p* < 0.01, *n* = 10). It was 0.34 ± 0.12 pA/ms with 30 μM 2-APB, compared to 0.24 ± 0.02 pA/ms without 2-APB (Figure 5D). The data suggested that 2-APB delayed the K_V_1.4 activation at a 2 ms length of the prepulse.

### 2.3. Effects of 2-APB on Mutation hKv1.4 Channels

#### 2.3.1. Mutations at the Pore Region

Considering that 2-APB exerts a much higher potency against K_V_1.4 channel currents, we further studied its inhibitory mechanism. Most known small-molecule inhibitors of K_V_ channels bind a water-filled cavity below the selectivity filter that is formed by residues located at the base of the selectivity filter and by pore-lining amino acids of the inner (S6) helices [13]. We hypothesized that 2-APB might have a specific interaction site(s) on the K_V_1.4 channel and that the 2-APB interaction site(s) with the K_V_1.4 channel might locate in the channel cavity region. To identify the site(s), we conducted alanine scanning by making a series of site mutations, except for the A553 site that was mutated to A553V, in the channel cavity region (G548A, V549A, L550A, T551A, I552A, A553V, L554A, P555A, V556A, P557A, V558A, I559A, V560A). These mutations were transiently expressed in CHO cell lines. Despite some of these mutations having no currents or too few currents elicited for analysis, 100 μM 2-APB was used for studying the mutations of measurable currents. We found that four mutations (V549A, T551A, A553V, and L554A) significantly attenuated the 2-APB inhibition of the K_V_1.4 channel currents (Figure 6), whereas the other mutations had no significant effects (Table 1). This indicated that 2-APB might regulate K_V_1.4 channel activity through interactions with the four residues.

#### 2.3.2. N-Terminal Truncation

K_V_1.4, as a typical A-type channel, has the fastest kinetic inactivation among K_V_1s, and 2-APB also has the strongest inhibition on K_V_1.4. It is interesting to investigate whether there is an impact on the inhibition of 2-APB if the fast inactivation is removed. Because the N-terminal of K_V_1.4 serves as the “ball” to deliver the fast inactivation by blocking the path of ions at the bundle cross of the S6 region [14], 1–61 amino acids at the N-terminal were removed to make a truncation mutation. Compared with the other mutations, the N-terminal-removed mutation was the least inhibited by 2-APB. Furthermore, 10, 30, 100, 300, and 1000 μM 2-APB inhibited peak amplitudes of K_V_1.4 currents by 1.5 ± 1.7%, 9.0 ± 0.9%, 23.1 ± 2.5%, 49.1 ± 2.9%, and 77.1 ± 3.8%, respectively (*n* = 5–6) (Figure 7). The IC_50_ value was 310.4 ± 16.6 μM (*n* = 6).

As the K_V_1.4 channels do not inactivate yet at 2 ms (Figure 5A,B), the activation delay is more probably correlated with the fast inactivation. Thus, the onset of the inactivation of the N-terminal truncation mutation was investigated. Since the N-terminal truncation mutation has lost inactivation, there was no clear inactivation detected in the first 32 ms (Figure 8A). Clear facilitation was discovered at at least 2, 4, and 8 ms with or without 2-APB. The facilitation of current amplitudes (I_2_/I_1_) at a 2 ms length prepulse was 2.23 ± 0.32 vs. 4.68 ± 0.69 before and after 30 μM 2-APB application (*p* < 0.05, *n* = 4) (Figure 8B). Similar to the effect on WT, the facilitation was more than doubled after 2-APB application. The average time needed to activate half of the maximal current was delayed by 0.4 ± 0.09 ms by 30 μM 2-APB (*p* < 0.05, *n* = 4). Meanwhile, the slope was 0.33 ± 0.02 pA/ms with 30 μM 2-APB, compared to 0.42 ± 0.07 pA/ms without 2-APB (*n* = 4, *p* > 0.05) (Figure 9).

The data suggested that the truncation of N-terminal reduced the 2-APB inhibition on K_V_1.4 channels but does not affect the short prepulse facilitation.

### 2.4. Docking of the Interactions between 2-APB and Three Channels

To further examine the possible interaction at the atomic level between the ligand 2-APB and hK_V_1.3 and hK_V_1.4 and the template structure of rK_V_1.2 channels, homology modeling of hK_V_1.3 and hK_V_1.4 and molecular docking of protein–ligands complexes were carried out. The sequence of the central pore of hK_V_1.3 and hK_V_1.4 was aligned with that of rK_V_1.2 according to the previous studies [15,16,17]. The hK_V_1.3, hK_V_1.4, and rK_V_1.2 share a ~90% sequence identity in the pore domain. Hence, using a Swiss-model workspace, we constructed a reliable 3D structure of hK_V_1.3 and hK_V_1.4 composed of S5, S6, and P-loop according to the crystal structure of rK_V_1.2. The quality of the refined models was then evaluated by the Ramachandran plot, which suggests that the Phi/Psi angles of most residues (~90%) are within the reasonable ranges. The flexible docking protocol was carried out to explore the molecular basis of the interaction between rK_V_1.2, hK_V_1.3, and hK_V_1.4 and 2-APB. Based on the docking analysis, the phenolic hydroxyl group of 2-APB showed an interaction with T401 (K_V_1.2 (2A79)), A473 (K_V_1.3 model), and V549 (K_V_1.4 model), respectively (Figure 10, Figure 11 and Figure 12). The best binding poses were chosen based on the binding energy score, the inhibition constant, the VDW_HB desolv_energy, and the ligand efficiency, and the surrounding residues of ligands may have an essential impact on the binding affinity between hK_V_1.4 and 2-APB. According to the computational docking results, the superposition of the binding structures for the inspected ligand is shown in Figure 10, Figure 11 and Figure 12. The inhibition constants (Ki) predicted were 218.87 μM, 350.52 μM, and 118.52 μM for rK_V_1.2, hK_V_1.3, and hK_V_1.4, respectively (Table 2). This shows a similar trend with the experimental data.

## 3. Discussion

Our study showed that 2-APB could significantly block three representative human K_V_1 channels. The most potent inhibition against A-type K_V_1.4 currents and a relatively weaker inhibition against two delayed outward rectifier channels were discovered. The mutations V549A, T551A, A553V, L554A, and N-terminal removal significantly attenuated 2-APB’s effects on K_V_1.4 channel currents. 2-APB did not affect the onset and recovery of K_V_1.4 inactivation but significantly slowed the initial activation kinetics of the channels. The importance of fast inactivation gating in determining the different 2-APB effects on the two types of channels was revealed. Interestingly, a current facilitation phenomenon by short prepulses was discovered for the first time.

Functional K_V_1 channels not only exist throughout both central and peripheral nervous systems but can also be expressed in peripheral tissues, including the cardiovascular and the immune system. Among them, K_V_1.2 is mostly in the central nervous system, and K_V_1.3 is more prominent in peripheral tissues. K_V_1.4, an A-type K_V_ channel, is widely distributed in excitable cells of mammalian tissues and exists in the cardiac ventricular endocardium [18,19]. The inhibitory effects of 2-APB on three channels can be used to explain the effects of 2-APB on the human body and animal models of human disease. Inhibition on K_V_1.3 ought to be a reason for modulating adaptive and innate immunity by 2-APB, because K_V_1.3, as a drug target for autoimmune diseases, widely exists in immune cells. Inhibition on K_V_1.2 and K_V_1.4 should play a role in 2-APB’s effects on neurons, smooth muscle cells, and cardiomyocytes [2]. Inhibition on both the transient and sustained voltage-activated potassium current of *Limulus ventral* photoreceptors comprising a delayed outward rectifier K^+^ current and a rapidly inactivating conductance could also result from 2-APB’s effects on Kv1 channels [7].

2-APB showed the most potent effects on K_V_1.4 among three channels. Previous studies showed that 2-APB’s inhibitory effects on K_V_ channels of guinea pig arteriole cells were more marked on the fast component than they were on the slow component [8]. Our results could provide an explanation.

Among the antagonists of K_V_1.4 channels, 2-APB, with an IC_50_ of 67.3 μM, is comparable to the effects of fluoxetine and La^3+^, and its inhibition is 4 and 100 times more potent than 4-aminopyridine and TEA, respectively [18]. The data indicated that the inhibitory potential of 2-APB is quite considerable. 2-APB has been widely used as a tool in many studies. It is worth noticing that 2-APB has also been used in several systems (e.g., rat dorsal root ganglia) that endogenously express K_V_1 as their major potassium background channels [20]. We think that any application of 2-APB over 10 μM at cellular levels should take into consideration the potential inhibition of K_V_1.4.

The V549A, T551A, A553V, and L554A mutations significantly attenuate 2-APB’s effects on K_V_1.4, and these four sites were conserved with K_V_1.2 (V399, T401, A403, L404) and K_V_1.3 (V469, T471, A473, L474) channels. It is reasonable to speculate that these sites might also play a key role in the inhibition of K_V_1.2 and K_V_1.3 by 2-APB. The variable IC_50_s in K_V_1.2 and K_V_1.3 indicated that other diverse regions or sites of these channels might be involved in the interaction between 2-APB and channels.

Fast inactivation is the characteristic of K_V_1.4 that distinguishes it from other K_V_1.x channels [21]. N-terminal removal will disrupt the fast inactivation of the channel. Our data of the N-terminal deleted channel showed the weakest inhibition by 2-APB, suggesting that the fast inactivation plays a vital role in the inhibition of 2-APB. Interestingly, N-terminal truncation caused the IC_50_ on K_V_1.4 to be nearly the same as that of K_V_1.2. Moreover, A553V mutation also disrupted the fast inactivation kinetics (Figure 6C) and mostly significantly attenuated 2-APB inhibition, which is consistent with the N-terminal truncation data. These findings provide additional evidence on the key role of the fast inactivation gate in the 2-APB effects on K_V_1.4. Considering that fast inactivation is the major difference between K_V_1.4 and the other K_V_1 channels, the fast inactivation characteristic of K_V_1.4 might determine the different inhibitory potencies of 2-APB against K_V_1.4 and the other two K_V_1channels.

Our data showed that 2-APB delayed the kinetic activation of K_V_1.4 when the prepulse length was 2 ms. The N-terminal truncated mutation altered neither this effect nor the amount of time delayed. We do not know whether other K_V_1.4 inhibitors have the same effect, or, in other words, whether this effect is 2-APB-specific. We suspect that 2-APB might have a direct impact on the channel opening, but further evidence is needed.

In the docking analysis, 2-APB had an interaction with V549 and did not bind with the other three key residues in the K_V_1.4 channel. 2-APB could also interact with T401 in Kv1.2 and A473 in K_V_1.3, which are conserved with T551 and A553 of the K_V_1.4 channel. The docking predictions were consistent with the inhibitory effects on three channels by 2-APB.

## 4. Materials and Methods

### 4.1. Materials

2-APB was purchased from Sigma (San Diego, CA, USA). The cDNAs for the human K^+^ channels K_V_1.2(NM_004974.4), K_V_1.3(NM_002232.5), and K_V_1.4 (NM_002233.4) were bought from OriGene Technologies (Rockville, MD, USA), subcloned into pcDNA3.1(neo^+^), and stably transfected into CHO cells, which were used to determine the IC_50_s of 2-APB in different channels.

### 4.2. Site-Directed and N-Terminal Truncation Mutations of K_V_1.4

Mutation PCR was performed with primers synthesized by Sangon biotech (Shanghai, China) using the KOD DNA polymerase (TOYOBO, Osaka, Japan). PCR products were digested by the DpnI restriction enzyme (TAKARA, Dalian, China), purified, and transformed into DH5α *E. coli* competent cells. Additionally, 1–61 amino acids were deleted for N-terminal truncation mutation. After sequencing, plasmids with the desired mutations were extracted and transiently transfected into Chinese hamster ovary (CHO) cells using Lipofectamine™ 2000.

### 4.3. Data Recording

CHO cells with K_V_1.2, 1.3, or 1.4 stably expressed were routinely cultured in the solution with 10% fetal bovine serum (FBS), 1% P/S (100U Penicillin and 0.1 mg/mL streptomycin), and 100 μg/mL G418. Cells not connected with other cells were voltage-clamped using the PC505B (Warner Instrument Corporation) patch clamp amplifier in the whole-cell configuration. Electrodes ranged from 2 to 3 MΩ in resistance. The voltage clamp data were filtered at 2 kHz and digitized at 100 or 150 ms/point. Voltage protocols were generated and analyzed by Clampex and Clampfit patch clamp software (Version 10.4, Axon Instruments). The recordings from cells were carried out at room temperature (25 ± 1 °C) [22].

### 4.4. Statistical Analysis

Electrophysiological data were analyzed by Clampfit software (Version 10.4, Axon Instruments) and Origin 8.0 (OriginLab Corporation, Northampton, MA, USA). All data were expressed as the mean ± S.E.M using Student’s *t*-tests with statistical significance (*p* < 0.05). The IC_50_ value was obtained by fitting the concentration-dependent data to the following Hill equation: I (%) = 1/{1 + (IC_50_/[D])^n^}. In the equation, I% is the percentage inhibition of current amplitudes; IC_50_ is the concentration of the half-maximal inhibition; [D] is the concentration of a compound; and n is the Hill coefficient.

### 4.5. Homology Modeling

The pore region sequences of hK_V_1.3 and hK_V_1.4 were retrieved from the Uniprot database (accession number: hK_V_1.3—P22001, hK_V_1.4—P22459), and multiple sequence alignment was carried out with the sequence of rat K_V_1.2. Since the crystal structure of the human voltage-gated potassium channels K_V_1.3 and K_V_1.4 has not yet been determined, the homology model of both channels was constructed by the Swiss-model based on the crystal structure of the rat K_V_1.2 channel (PDB entry: 2A79) [14,23]. The residues Met288 to Thr421, forming the S5 pore helix and S6 in rat K_V_1.2, correspond to the region from Met358 to Thr491 in hK_V_1.3, and Met438 to Thr571 in hK_V_1.4 was selected as the template for homology modeling. The sequence identity of these regions between the rat K_V_1.2, hK_V_1.3, and hK_V_1.4 was ~90%, which can enable us to construct a reliable homology model based on the high-resolution crystal structure of rat K_V_1.2. As the crystal structure of rat K_V_1.2 contains only one subunit, the transformation matrices of the structural coordinate file were employed to generate the missing subunits of rat K_V_1.2, creating the fourfold symmetry required to build the hK_V_1.3 and hK_V_1.4 channel homo-tetramer. The Ramachandran plot further assessed the quality of the refined model.

### 4.6. Induced Fit Docking

The compound 2-APB, the modeled target proteins hK_V_1.3 and hK_V_1.4, and the template structure of rat K_V_1.2 were used in the molecular docking analysis by the AutoDock tools (ADT) with the MGL Tools v1.5.6rc3 program [24]. All three target protein molecules were prepared by the Python Molecule Viewer (PMV); after repairing the missing atoms, the polar hydrogen atoms and Gasteiger charges were added into the protein structures. The active binding sites were selected at the central cavity of the pore region of hK_V_1.3 and hK_V_1.4 in each of the four subunits and were considered as flexible according to the previous study [25]; the rest of the target protein sites were treated as a rigid body. The parameter library was loaded for the ligand molecule 2-APB, and the atoms were also considered flexible. The grid map was constructed based on the ligand atom types with a default grid spacing of 0.375 A° in the box size of 90 × 90 × 90. After the grid map calculation, the target proteins and the ligand molecule were induced for docking analysis. The default genetic algorithm parameters with the addition of AutoDock 4.2 parameters were used for building 27,000 generations with a population size of 150 individuals. The docking conformations were calculated by the Lamarckian genetic algorithm (LGA) [26]. Finally, the least scored conformation was evaluated and picked for the ligand-receptor docking analysis; the interaction images were developed by PyMOL.

## 5. Conclusions

In conclusion, this study demonstrated that 2-APB could significantly block three representative human K_V_1 channels. Moreover, the potential inhibitory mechanism was also investigated. Therefore, 2-APB’s effects on K_V_1 channels might be part of the reason for the diverse bioactivities of 2-APB in the human body and in animal models of human disease. In other words, our study revealed new targets of 2-APB at the molecular level. More comprehensive research is needed to understand the inhibitory mechanism in more detail and more 2-APB effects on other channels.

## Figures and Tables

**Figure 1 molecules-28-00871-f001:**
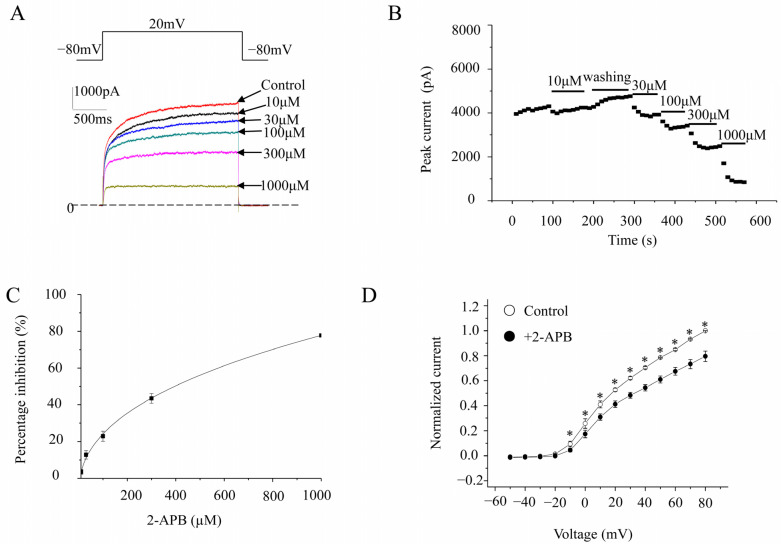
Concentration-dependent inhibition of K_V_1.2 channels. (**A**). Representative K_V_1.2 channel currents elicited by 2000 ms voltage steps up to +20 mV from a holding potential of −80 mV, in the absence and presence of different concentrations of 2-APB. (**B**). The time course of K_V_1.2 currents after 2-APB treatment. (**C**). Concentration response curve of K_V_1.2 currents inhibited by 2-APB. The percentage inhibitions against the 2-APB concentration were analyzed by the Hill equation. (**D**). I-V relationships of wild-type K_V_1.2 channels in the absence or presence of 2-APB. Voltage pulses 2000 ms in duration were applied from −50 mV to 80 mV with 10 mV increments each time, with a holding potential of −80 mV. The interval between pulses was 10 s. * represents *p* < 0.01.

**Figure 2 molecules-28-00871-f002:**
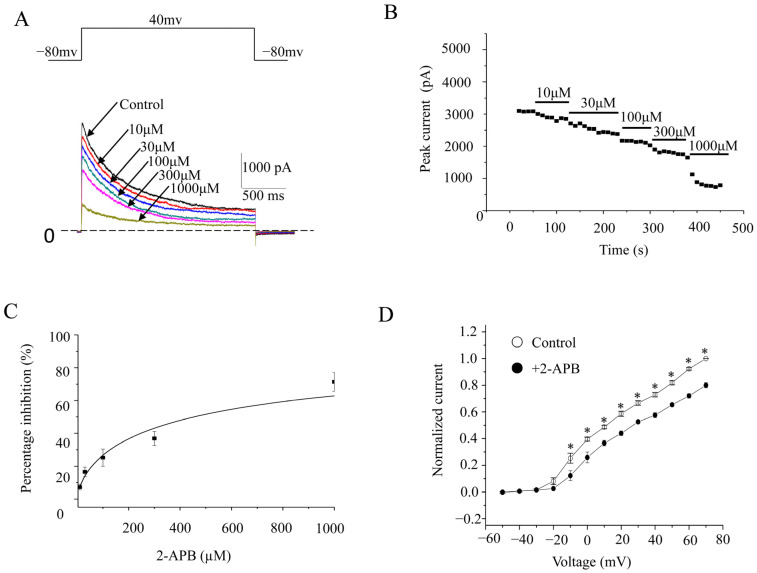
Concentration-dependent inhibition of K_V_1.3 channels. (**A**). Representative K_V_1.3 channel currents elicited by 2000 ms voltage steps up to +40 mV from a holding potential of −80 mV, in the absence and presence of different concentrations of 2-APB. (**B**). The time course of K_V_1.3 currents treated with different 2-APB concentrations. (**C**). Concentration response curve of hK_V_1.3 currents inhibited by 2-APB. The percentage inhibitions against the 2-APB concentration were analyzed by the Hill equation. (**D**). I-V relationships of wild-type K_V_1.3 channels in the absence or presence of 2-APB. Voltage pulses 2000 ms in duration were applied in 10 mV increments and at 10 s intervals from a holding potential of −80 mV. * represents *p* < 0.01.

**Figure 3 molecules-28-00871-f003:**
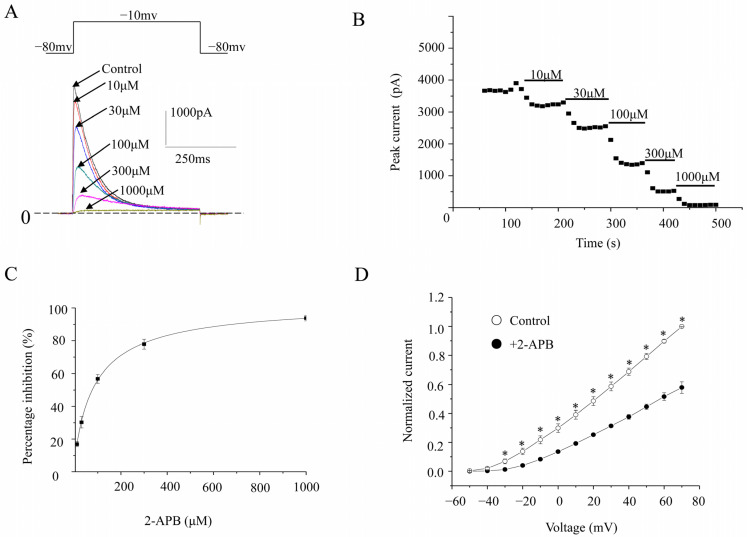
Concentration-dependent inhibition of K_V_1.4 channels. (**A**). Representative K_V_1.4 channel currents elicited by 500 ms voltage steps up to −10 mV from a holding potential of −80 mV, in the absence and presence of different concentrations of 2-APB. (**B**). The time course of hK_V_1.4 currents after 2-APB treatment. (**C**). Concentration response curve of K_V_1.4 currents inhibited by 2-APB. The percentage inhibition against the 2-APB concentration was analyzed by the Hill equation. (**D**). I-V relationships of wild-type K_V_1.4 channels in the absence or presence of 2-APB. Voltage pulses 500 ms in duration were applied in 10 mV increments and at 10 s intervals from a holding potential of −80 mV. * represents *p* < 0.01.

**Figure 4 molecules-28-00871-f004:**
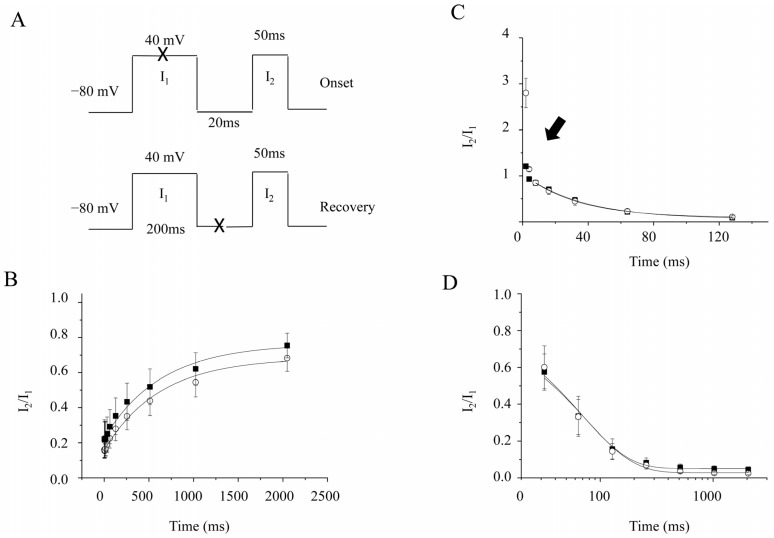
Effects of 2-APB on the onset and recovery of K_V_1.4 channels. ○ represents data after 2-APB application. ■ represents data before 2-APB application. I_1_ and I_2_ are the peak current amplitudes of the prepulse and test pulse. (**A**). **Top panel**: protocol for studying the onset of inactivation. **Bottom panel**: protocol for studying the recovery of inactivation. “X” means various lengths of duration. (**B**). The recovery of wild-type K_V_1.4 channels in the absence or presence of 2-APB. The X-axis represents the length between the prepulse and the test pulse. (**C**). The onset of the fast inactivation of K_V_1.4 channels in the absence or presence of 2-APB. The prepulse length varies from 2 ms to 128 ms. 
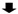
 points to the current amplitudes facilitated at a 2 ms prepulse length. (**D**). The onset of the slow inactivation of K_V_1.4 channels in the absence or presence of 2-APB. The prepulse length varied from 32 ms to 1024 ms.

**Figure 5 molecules-28-00871-f005:**
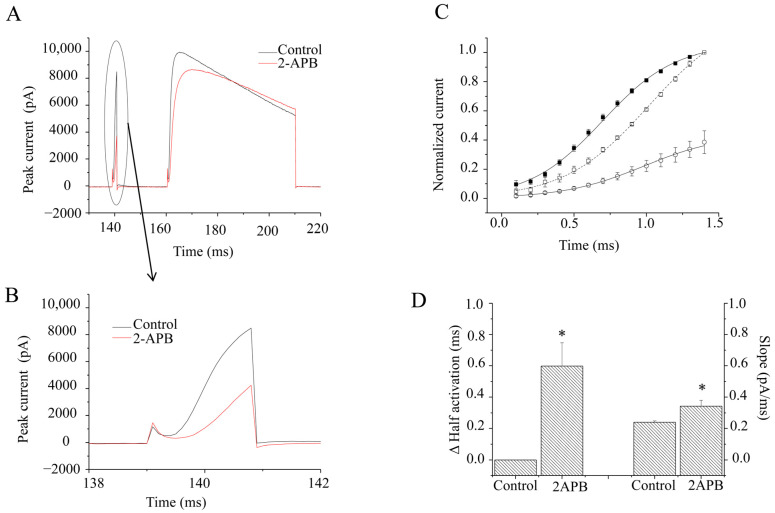
Effects of 2-APB (30 µM) at a 2 ms prepulse at the onset. (**A**). Example current traces with (Black) and without 2-APB (Red). (**B**). Enlarged example prepulse current traces with (Black) and without 2-APB (Red). (**C**). Averages of current amplitudes at different times were analyzed by the Boltzman equation. ○ represents data after 2-APB application. ■ represents data before 2-APB application. Current amplitudes were normalized to the maximal peak currents in prepulses without 2-APB. □ represents the normalized current amplitudes with 2-APB, which were normalized to the maximal peak currents in the prepulse with 2-APB. (**D**). Bar graphs show the half-current activation time (**left**), which is relative to that without 2-APB treatment, and slopes (**right**) with and without 2-APB. * represents *p* < 0.01.

**Figure 6 molecules-28-00871-f006:**
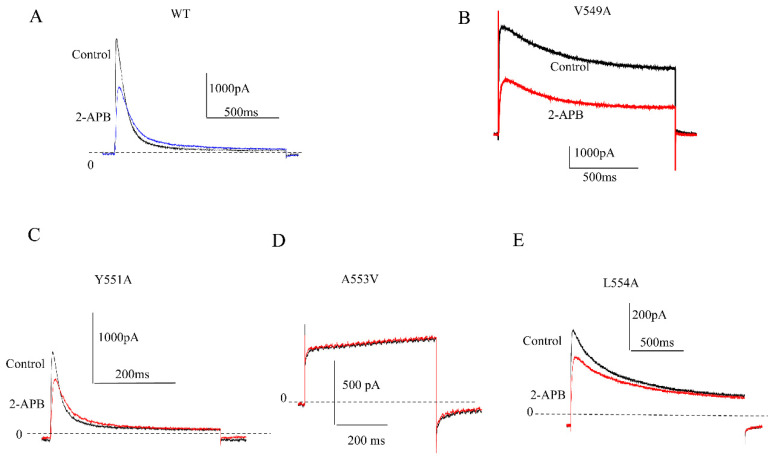
Effects of 100 μM 2-APB on mutations of the K_V_1.4 channel. (**A**). Representative wild-type currents elicited in the absence and presence of 2-APB. (**B**). Representative V549A currents elicited in the absence and presence of 2-APB. (**C**). Representative T551A currents elicited in the absence and presence of 2-APB. (**D**). Representative A553V currents elicited in the absence and presence of 2-APB. (**E**). Representative L554A currents elicited in the absence and presence of 2-APB.

**Figure 7 molecules-28-00871-f007:**
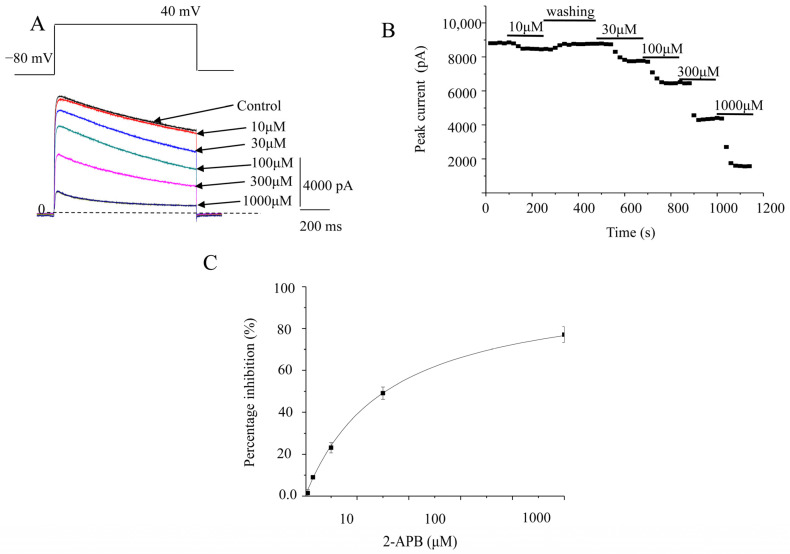
Effects of 2-APB on the N-terminal truncated mutation of K_V_1.4. (**A**). Representative N-terminal truncated mutation of K_V_1.4 channel currents elicited in the absence and presence of various concentrations of 2-APB. (**B**). The time course of 2-APB effects on the N-terminal truncated mutation of K_V_1.4 currents (100 μM). (**C**). Concentration response curve of the N-terminal truncated mutation of K_V_1.4 channel currents inhibited by 2-APB. The percentage inhibitions against the 2-APB concentration were analyzed by the Hill equation.

**Figure 8 molecules-28-00871-f008:**
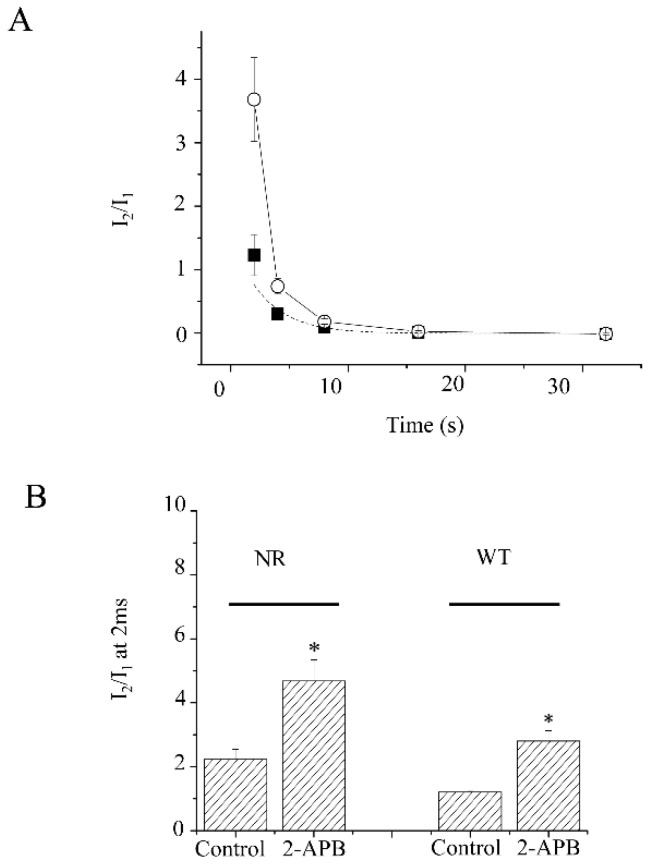
The fast onset of the N-terminal truncated mutation of K_V_1.4. (**A**). The onset of the fast inactivation of the N-terminal truncated mutation of K_V_1.4 channels in the absence or presence of 2-APB. The prepulse length varied from 2 ms to 32 ms. The protocol was the same as that in Figure 4A. Current amplitudes are facilitated at a 2 ms prepulse length. (**B**). The bar graph shows the facilitation of both the N-terminal truncated K_V_1.4 and wild-type-K_V_1.4 with and without 2-APB at a 2 ms prepulse. ■ represents the data of wild-type K_V_1.4. ○ represents the data of N-terminal truncated K_V_1.4. * represents *p* < 0.05.

**Figure 9 molecules-28-00871-f009:**
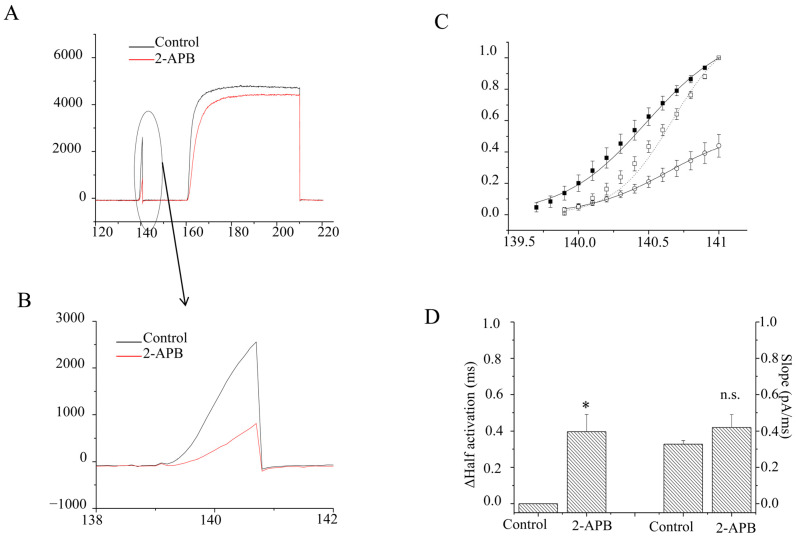
Effects of 2-APB at a 2 ms prepulse at the onset of the N-terminal truncated mutation of K_V_1.4. (**A**). Example current traces with (Black) and without 2-APB (Red). (**B**). Enlarged example prepulse current traces with (Black) and without 2-APB (Red). (**C**). Average current amplitude at different times analyzed by the Bolzman equation. ○ represents data after 2-APB application, and ● represents data before 2-APB application. Current amplitudes were normalized to the maximal peak currents in prepulses without 2-APB. □ represents the normalized current amplitudes with 2-APB, which were normalized to the maximal peak currents in a prepulse with 2-APB. (**D**). Bar graphs show the half-current activation time (**left**), relative to the one without 2-APB, and a slope (**right**) with and without 2-APB. * represents *p* < 0.05; “n.s.” means “not significant”.

**Figure 10 molecules-28-00871-f010:**
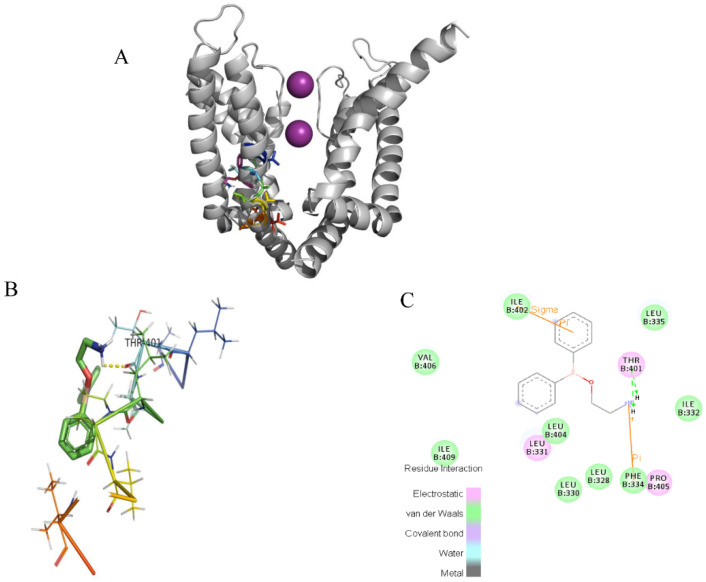
Induced-fit docking model of 2-APB binding K_V_1.2. (**A**). Side (**right**) views of the docked ligands in complex with the K_V_1.2 model. (**B**). Stereoscopic view of 2-APB binding active sites of K_V_1.2. (**C**). Planar view of 2-APB binding active sites of K_V_1.2. “----” shows the hydrogen bond between 2-APB and T401.

**Figure 11 molecules-28-00871-f011:**
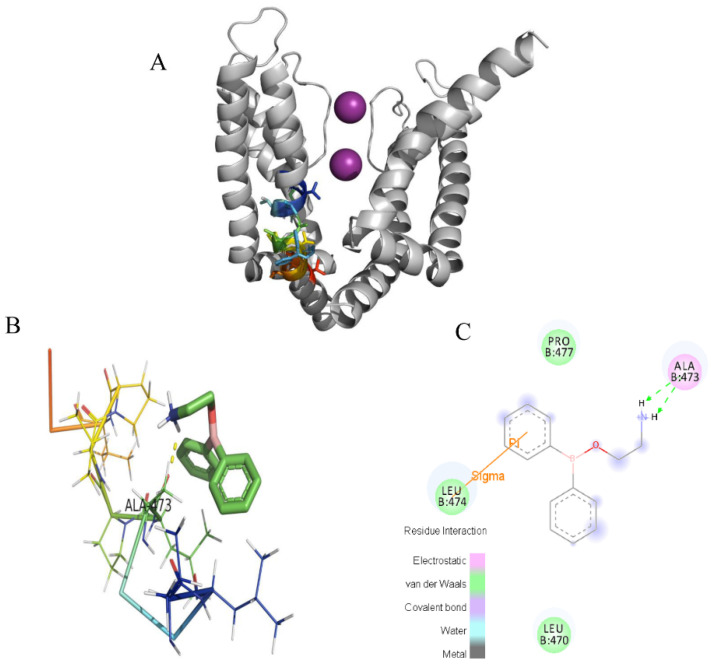
Induced-fit docking model of 2-APB binding K_V_1.3. Induced-fit docking model of 2-APB binding K_V_1.3. (**A**). Side (**right**) views of the docked ligands in complex with the K_V_1.3 model. (**B**). Stereoscopic view of 2-APB binding active sites of K_V_1.3. (**C**). Planar view of 2-APB binding active sites of K_V_1.3. “----” shows the hydrogen bond between 2-APB and A473.

**Figure 12 molecules-28-00871-f012:**
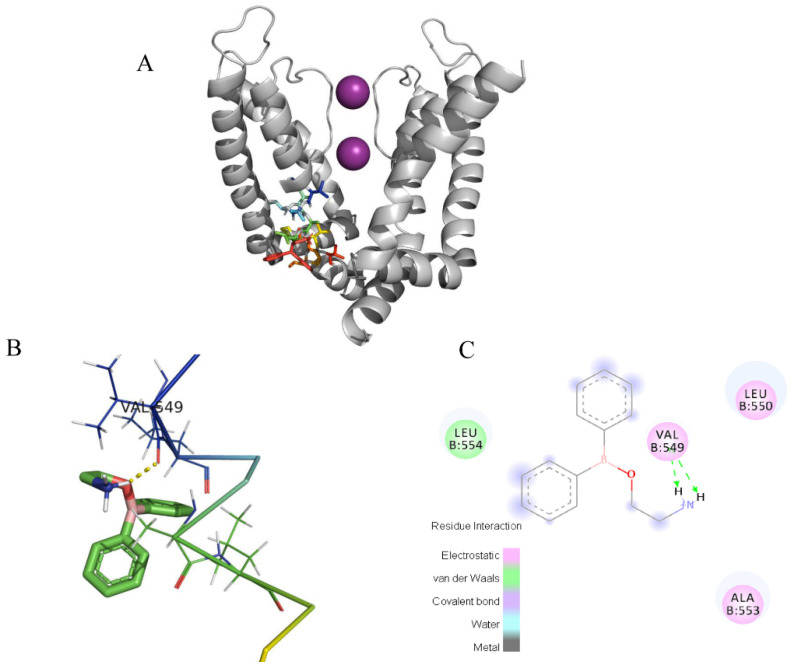
Induced-fit docking model of 2-APB binding K_V_1.4. Induced-fit docking model of 2-APB binding K_V_1.4. (**A**). Side (**right**) views of the docked ligands in complex with the K_V_1.4 model. (**B**). Stereoscopic view of 2-APB binding active sites of K_V_1.4. (**C**). Planar view of 2-APB binding active sites of K_V_1.4. “----” shows the hydrogen bond between 2-APB and V549.

**Table 1 molecules-28-00871-t001:** Effects of 2-APB (100 μM) on mutation Kv1.4 channels expressed in CHO cells.

Mutations	Inhibition (%)	S.E.M	Number of Cells
WT	56.8	2.7	6
K_V_1.4_G548A	N.D.	N.D.	N.D.
K_V_1.4_V549A	44.1 *	0.2	4
K_V_1.4_L550A	50.0	4.4	4
K_V_1.4_T551A	37.8 *	6.0	4
K_V_1.4_I552A	N.D.	N.D.	N.D.
K_V_1.4_A553V	18.4 *	7.0	6
K_V_1.4_L554A	32.9 *	9.7	4
K_V_1.4_P555A	N.D.	N.D.	N.D.
K_V_1.4_V556A	60.8	0.5	3
K_V_1.4_P557A	N.D.	N.D.	N.D.
K_V_1.4_V558A	57.3	4.1	5
K_V_1.4_I559A	N.D.	N.D.	N.D.
K_V_1.4_V560A	N.D.	N.D.	N.D.
K_V_1.4_S561A	N.D.	N.D.	N.D.
K_V_1.4_ truncation	23.1 *	2.5	5

N.D., not determined. * *p* < 0.01 compared with the wild-type K_V_1.4 channel.

**Table 2 molecules-28-00871-t002:** Docking parameters of 2-APB binding three channels.

Ligand	Protein	Binding Amino Acid Residues	Binding Energy (kcal/mol)	Inhibition Constant, Ki (μM)	VDW_HB Desolv_Energy (kcal/mol)	Ligand Efficiency
2-APB	Kv1.2 (2A79)	T401	−6.04	218.87	−6.00	0.36
	Kv1.3 (model)	A473	−5.56	350.52	−6.56	0.30
	Kv1.4 (model)	V549	−7.26	118.52	−7.56	0.43

## Data Availability

Not applicable.

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
