# Peer review of "Inhibitory Effects of 2-Aminoethoxydiphenyl Borate (2-APB) on Three KV1 Channel Currents"

_molecules, 2023, doi:10.3390/molecules28020871_

Round 1
Reviewer 1 Report
I have no comments on the manuscript except that all the figures need to be improved.
Author Response
Reviewer 1: I have no comments on the manuscript except that all the figures need to be improved.
Answer: Thank you very much for the comments. The figure layout and sharpness have been improved in the new version.
Reviewer 2 Report
Zhao et al. studied the effects of a boron-conteining molecule (2-APB) on the currents of the Kv1.2, Kv1.3 and Kv1.4 channels; In addition, through molecular docking, they evidenced the potential interactions of this compound with these channels. In general, the manuscript is well written and follows a logical sequence. The electrophysiological analysis is interesting, well presented and understandable. The results could be important in translational research.
However, prior to publication, I would appreciate it if the authors could address the following points in the manuscript:
- Why were the I-V curves made with 100 µM of 2-APB and not with their respective IC50 value? Likewise, could point-to-point statistical significance be added to this type of curves (Figure 1D, 2D and 3D) if it existed?
- Line 171: Was the voltage pulse +30 mV as claimed in the figure description? Correct as shown in Figure 2A if applicable.
- In Figure 4, the effects of 2-APB on initiation and recovery of Kv1.4 channels are shown. The description of this figure refers to the symbol "●" to refer to the effect in the absence of 2-APB, however the symbol "■" is used in the figures. Verify and correct also for the other figures if necessary.
- Although it is mentioned in the results section, in Figure 5 it should be specified that 30 µM of 2-APB was used (in previous experiments the authors used 100 µM). This is so that the figure can be understood without the need to resort to the full text. Likewise, it should be described what the symbol * means: p<0.05, p<0.01, p<0.001?
- In figure 8B, the symbol of significant difference between both bars must be shown if it exists (n.s.= not significant, if there is no significance). The same for figure 9D.
- Present the abstract without explicitly writing the headings.
- Authors should check the journal's instructions regarding bibliographic references and how they should be presented in the text of the manuscript.
